# Establishment of transgenic fluorescent mice for labeling synapses and screening synaptogenic adhesion molecules

Lei Yang[1†], Jingtao Zhang[1†], Sen Liu[1†], Yanning Zhang[1†], Li Wang[1†], Xiaotong Wang[1], Shanshan Wang[2], Ke Li[1], Mengping Wei[1*], Chen Zhang[1,3*]

[1]School of Basic Medical Sciences, Beijing Key Laboratory of Neural Regeneration and Repair, Advanced Innovation Center for Human Brain Protection, Capital Medical University, Beijing, China; [2]Peking-Tsinghua Center for Life Sciences, Academy for Advanced Interdisciplinary Studies, Peking University, Beijing, China; [3]Chinese Institute for Brain Research, Beijing, China

**\*For correspondence:**
weimengping@ccmu.edu.cn
(MW);
czhang@ccmu.edu.cn (CZ)

[†]These authors contributed equally to this work

**Competing interest:** The authors declare that no competing interests exist.

**Abstract** Synapse is the fundamental structure for neurons to transmit information between cells. The proper synapse formation is crucial for developing neural circuits and cognitive functions of the brain. The aberrant synapse formation has been proved to cause many neurological disorders, including autism spectrum disorders and intellectual disability. Synaptic cell adhesion molecules (CAMs) are thought to play a major role in achieving mechanistic cell-cell recognition and initiating synapse formation via trans-synaptic interactions. Due to the diversity of synapses in different brain areas, circuits and neurons, although many synaptic CAMs, such as Neurexins (NRXNs), Neuroligins (NLGNs), Synaptic cell adhesion molecules (SynCAMs), Leucine-rich-repeat transmembrane neuronal proteins (LRRTMs), and SLIT and NTRK-like protein (SLITRKs) have been identified as synaptogenic molecules, how these molecules determine specific synapse formation and whether other molecules driving synapse formation remain undiscovered are unclear. Here, to provide a tool for synapse labeling and synaptic CAMs screening by artificial synapse formation (ASF) assay, we generated synaptotagmin-1-tdTomato (*Syt1*-tdTomato) transgenic mice by inserting the tdTomato-fused synaptotagmin-1 coding sequence into the genome of C57BL/6J mice. In the brain of *Syt1*-tdTomato transgenic mice, the tdTomato-fused synaptotagmin-1 (SYT1-tdTomato) signals were widely observed in different areas and overlapped with synapsin-1, a widely-used synaptic marker. In the olfactory bulb, the SYT1-tdTomato signals are highly enriched in the glomerulus. In the cultured hippocampal neurons, the SYT1-tdTomato signals showed colocalization with several synaptic markers. Compared to the wild-type (WT) mouse neurons, cultured hippocampal neurons from *Syt1*-tdTomato transgenic mice presented normal synaptic neurotransmission. In ASF assays, neurons from *Syt1*-tdTomato transgenic mice could form synaptic connections with HEK293T cells expressing NLGN2, LRRTM2, and SLITRK2 without immunostaining. Therefore, our work suggested that the *Syt1*-tdTomato transgenic mice with the ability to label synapses by tdTomato, and it will be a convenient tool for screening synaptogenic molecules.

## Editor's evaluation

In this manuscript, Zhang and colleagues created a transgenic mouse strain that express SYT-1-tdt in all neurons. They showed that the labelled SYT-1 does not represent a strong over expression, colocalizes with multiple synaptic markers and label synapses in different regions. Importantly, they showed that the transgenic expression does not alter synaptic function using electrophysiogical assays. This reagent can be used to visualize synapses in vivo and in cultures.

## Introduction

Synapses, consisting of the presynaptic membrane, the synaptic cleft, and the postsynaptic membrane, are the intercellular connections that rapidly transmit information from presynaptic neurons to post-synaptic cells in a point-to-point manner (*Südhof, 2018*). Synapse formation is an essential stage in brain development that begins during embryonic/postnatal development, and lasts throughout life (*Biederer and Stagi, 2008*). The proper formation of synapses is crucial for neural circuits' formation and cognitive function development, and abnormal synapse formation can cause many neurological disorders, including autism spectrum disorders (ASD) and mental retardation (*Boda et al., 2010*; *McAllister, 2007*; *Südhof, 2008*; *Zhang et al., 2009*).

Synaptic CAMs are molecules that act as a key role in initiating synapse formation through trans-synaptic interactions, and were originally proposed to enable mechanistic cell-cell recognition (*Bukalo and Dityatev, 2012*; *Sanes and Yamagata, 2009*; *Yang et al., 2014*). A variety of CAMs have been shown to initiate synapse formation, anchor and organize the precise alignment of the pre- and postsynaptic sides of a synapse, and enable enhanced short- and long-term synaptic plasticity of synaptic transmission (*Südhof, 2018*; *Scheiffele et al., 2000*; *Graf et al., 2004*; *Linhoff et al., 2009*; *de Wit et al., 2009*; *Ko et al., 2009*; *Kim et al., 2006*; *Woo et al., 2009*; *Wang et al., 2021*; *Takahashi et al., 2011*; *Sando et al., 2019*; *Pettem et al., 2013*; *Yim et al., 2013*; *Biederer et al., 2002*). However, the synapses show diversities in types of neurotransmitter, release probability, and composition of postsynaptic receptors in different brain areas, and are organized by specific CAMs (*Takahashi et al., 2011*; *Südhof, 2021*; *Tanabe et al., 2017*; *Takahashi et al., 2012*). Deletion of some CAMs only affects the pre- or post-synaptic organization rather than affects synapse numbers (*Linhoff et al., 2009*; *Varoqueaux et al., 2006*). In other cases, deletion of specific CAMs only induces the loss of synapse number in specific type of neurons or brain areas (*Zhang et al., 2018*; *Chen et al., 2017*). Therefore, how the CAMs determine specific synapse formation still needs to be elucidated and it cannot be excluded that some other CAMs driving synapse formation remain undiscovered.

Up to now, a large-scale screen for synaptogenic CAMs remains difficult to implement. A study identifies gene transcripts encoding proteins expressed by postsynaptic neurons that potentially initiate the synaptic contacts' formation by performing a genome-wide, expression analysis of chick ciliary ganglion at the different stages of synapse formation (*Brusés, 2010*). This screening method is comparatively complex and does not allow for direct observation of synapse formation. Another method for screening CAMs is the artificial synapse formation (ASF) assay (*Scheiffele et al., 2000*). It works by co-culturing non-neuronal cells, such as HEK293T cells, which are overexpressed by certain genes, with neurons. It can be observed that the generated synapses wrap around non-neuronal cells when the gene overexpressed has the synaptogenesis function. Many synaptogenic CAMs have been found to accumulate synapses around the non-neuronal cells in ASF experiments, including Neurexins (NRXNs), Neuroligins (NLGNs), Synaptic cell adhesion molecules (SynCAMs), Leucine-rich-repeat transmembrane neuronal proteins (LRRTMs), Latrophilins, Protein tyrosine phosphatase receptor type O (PTPRO), and others (*Scheiffele et al., 2000*; *Graf et al., 2004*; *Linhoff et al., 2009*; *Biederer et al., 2002*; *Jiang et al., 2017b*; *Sando et al., 2019*; *Nam and Chen, 2005*), which indicate that ASF assay is suitable for identifying new synaptogenic CAMs. However, ASF assay still requires a lot of time for immunohistochemically staining, making it difficult to perform large-scale screening.

Synaptotagmin-1 (SYT1), a fast calcium sensor on synaptic vesicles (*Kochubey et al., 2016*; *Xu et al., 2007*), begins to be strongly expressed on embryonic day 18 and continues to increase until reaching stable levels in adulthood (postnatal day 60) in many brain regions (*Wolfes and Dean, 2020*). SYT1 is the most abundant synaptotagmin in synaptic vesicles (*Fukuda, 2006*). In the hippocampus, SYT1 is abundant in the stratum oriens, stratum radiatum, and stratum lacunosum molecular (*Fox and Sanes, 2007*). In this study, we fused tdTomato (tdTomato) to the end of the SYT1 protein, and used the PiggyBac transposon vector system to insert the synaptotagmin1-tdTomato (*Syt1*-tdTomato) gene into C57BL/6 J mice allete. The tdTomato signals were widely observed in the brain of *Syt1*-tdTomato transgenic mice, and colocalized with synaptic markers. We also observed the enrichment of SYT1-tdTomato in the glomerulus of the olfactory bulb. The transgenic mice can also be used for ASF experiments in order to visualize synapses by fluorescence microscopy without immunostaining. Therefore, the *Syt1*-tdTomato transgenic mice will be a valuable tool for large-scale screening of CAMs.

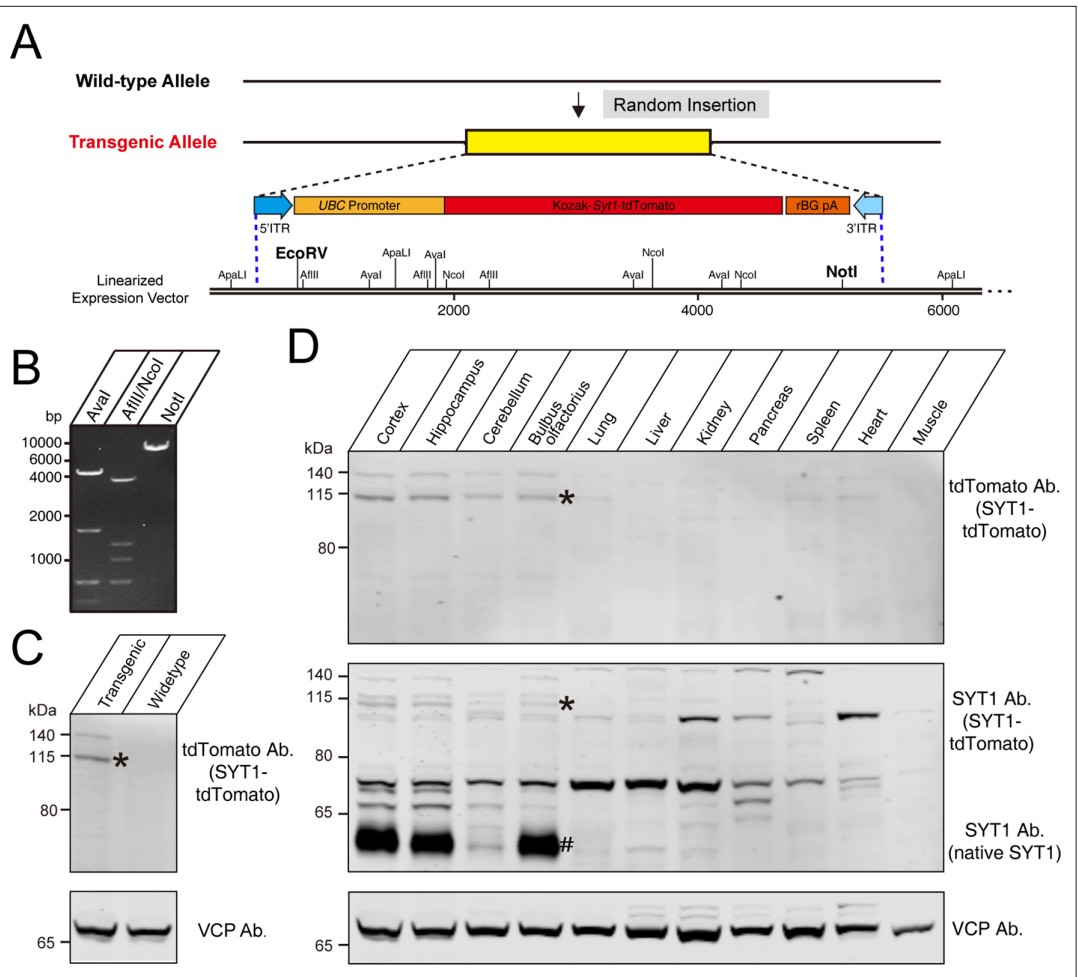

**Figure 1.** Construction of tdTomato-fused synaptotagmin-1 (*Syt1*-tdTomato) expression vector and transgenic fluorescent mice. (**A**) Schematic diagram of expression vector for generating *Syt1*-tdTomato transgenic mice. (**B**) The expression vector was digested by restriction enzymes for confirmation purposes. The size of enzyme digestion band (kb): (left) AvaI: 4.7/1.7/0.7/0.5; (middle) AflII/NcoI: 4.0/1.3/1.1/0.7/0.4/0.2; (right) NotI: 7.6. (**C**) Immunoblot analysis of protein in transgenic and wild-type mice brain. Equivalent amounts of protein were probed with antibodies to tdTomato and vasolin-containing protein (VCP, used as a loading control). (**D**) Immunoblot analysis of the expression of the transgenic protein in different tissues of *Syt1*-tdTomato transgenic mice. Equivalent amounts of protein were examined by immunoblotting with antibodies to tdTomato, SYT1, and VCP. Asterisk shows the band of SYT1-tdTomato and the hash shows the band of native SYT1.

The online version of this article includes the following source data for figure 1:

**Source data 1.** The original gel of panels B, C and D.

## Results

### *Syt1*-tdTomato transgenic mice express widely distributed fluorescent signals in synapse-rich regions of the brain

We used fertilized eggs microinjection with PiggyBac vectors carrying *Syt1*-tdTomato to generate a transgenic mouse line (C57BL/6 J background) with insertions of tdTomato-fused *Syt1* into a mouse allele (*Figure 1A*). In the PiggyBac vector, there were two PiggyBac ITRs on either side of the Human ubiquitin C (*UBC*) promoter-Kozak-*Syt1*-tdTomato-polyA in order to facilitate transposon-mediated transgene integration. To verify the successful construction of a *Syt1*-tdTomato expression vector, we digested the targeting vectors with restriction enzymes (*Figure 1A and B*). According to the schematic diagram of the linearized expression vector, the fragment sizes we obtained after enzyme digestion of the expression vectors should be AvaI: 4.7/1.7/0.7/0.5 kb (*Figure 1B*, left); AflII/NcoI:

4.0/1.3/1.1/0.7/0.4/0.2 kb (*Figure 1B*, middle); NotI: 7.6 kb (*Figure 1B*, right), which was consistent with our design. After a successful construction, the *Syt1*-tdTomato expression vector was co-injected with transposed into fertilized eggs from C57BL/6 J mice. The offspring that carry the desired PiggyBac transgene were identified and selected by polymerase chain reaction (PCR).

To verify the expression of *Syt1*-tdTomato in the transgenic mice, we analyzed the expression of the *Syt1*-tdTomato in the brains of transgenic mice by immunoblotting. Immunoblotting using antibodies against tdTomato detected bands in the whole brain homogenate of *Syt1*-tdTomato transgenic mice at 8 weeks postnatal, but not in control mice (*Figure 1C*). We next probed the tissue specificity of transgene expression (*Figure 1D*). Among the tissues examined, tdTomato was detectable in the cortex, hippocampus, cerebellum, and olfactory bulb of the *Syt1*-tdTomato transgenic mice brain, and faint bands were present in the heart homogenate at the same size position. To assess the overexpression level of SYT1-tdTomato, we immunoblot the SYT1-tdTomato and native SYT1 using antibody against SYT1. We found that the expression level of SYT1-tdTomato is far less than the native SYT1 in the brain of transgenic mice (*Figure 1D*), suggesting less effect of overexpressed SYT1-tdTomato on native SYT1's function.

To identify the distribution of SYT1-tdTomato signals in the brain of transgenic mice, we made frozen brain slices from 8-week-old *Syt1*-tdTomato transgenic mice and WT mice, and the brain slices were stained with DAPI to visualize the cell nucleus. We observed that SYT1-tdTomato was widely expressed in the hippocampus, cortex, cerebellum, and olfactory bulb in transgenic mice, but not in WT mice (*Figure 2A*). In the hippocampus of transgenic mice, the signals of SYT1-tdTomato were relatively bright in the stratum oriens and stratum radiatum of CA1-CA3 and stratum lucidum of CA3, while the signals were absent in the stratum pyramidale (*Figure 2B*). In the cerebellum, the signals were stronger in the molecular layer compared with the granular layer (*Figure 2C*). In the olfactory bulb, the SYT1-tdTomato showed bright signals in the glomerular layer (*Figure 2D*). Thus, our results suggested a wide distribution of SYT1-tdTomato in different brain areas, especially synapse-rich regions.

## SYT1-tdTomato signals colocalize with synaptic markers in situ and in cultured neurons

To observe whether the signals of SYT1-tdTomato localized well to the synaptic site, we performed immunostaining of synaptic markers in brain slices and cultured hippocampal neurons from *Syt1*-tdTomato transgenic mice. The signals of SYT1-tdTomato showed the same distribution as signals of synapsin-1 (SYN1), a widely-used synaptic marker, in hippocampus and cerebellum (*Figure 3A*). In the granular layer of the cerebellum and the glomerular layer of the olfactory bulb, the signals of SYT1-tdTomato were highly overlapped with signals of synapsin (*Figure 3B*). In addition, we quantified the intensity of SYT1-tdTomato signals in the olfactory bulb and found that the signals of SYT1-tdTomato were much higher in glomerular layer than that in the external plexiform layer (*Figure 3C*), suggesting an enrichment of SYT1-tdTomato in the glomerular layer of the olfactory bulb. These results suggest that SYT1-tdTomato could label synapse in situ in the brain, especially in the glomerular layer of the olfactory bulb.

In the cultured neurons, we could also observe strong tdTomato fluorescence which co-localized with the staining of SYN1 (*Figure 4A*). Furthermore, we performed the immunostaining of pre- and post-synaptic markers, including Vesicular glutamate transporter 1 (VGLUT1), Glutamic acid decarboxylase 65 (GAD65), Postsynaptic density protein 95 (PSD95), Homer protein homolog 1 (Homer-1), and Gephyrin. Our results showed that the SYT1-tdTomato signals co-localized well with these synaptic markers (*Figure 4B*). The co-localization rates were identified as percentage of signal A colocalized with signal B in total signal A (A-B): SYN1-tdTomato: 70.55 ± 1.447%; tdTomato-SYN1: 48.55 ± 1.851%; VGLUT1-tdTomato: 61.09 ± 1.517%; tdTomato-VGLUT1: 52.88 ± 1.456%; GAD65-tdTomato: 52.08 ± 2.442%; tdTomato-GAD65: 44.43 ± 2.096%; PSD95-tdTomato: 68.50 ± 2.397%; tdTomato-PSD95: 62.38 ± 2.433%; Homer-1-tdTomato: 61.00 ± 2.127%; tdTomato-Homer-1: 62.13 ± 1.703%; Gephyrin-tdTomato: 48.87 ± 2.466%; tdTomato-Gephyrin: 39.36 ± 1.665% (*Figure 4C*). These results indicate that SYT1-tdTomato signals localize well to the synaptic site in the neurons from transgenic mice.

## Neurons from *Syt1*-tdTomato transgenic mice show normal excitability and synaptic transmission

To further explore whether the insertion of *Syt1*-tdTomato affects the electrophysiological properties and synaptic neurotransmission of neurons in transgenic mice, we performed patch-clamp recordings

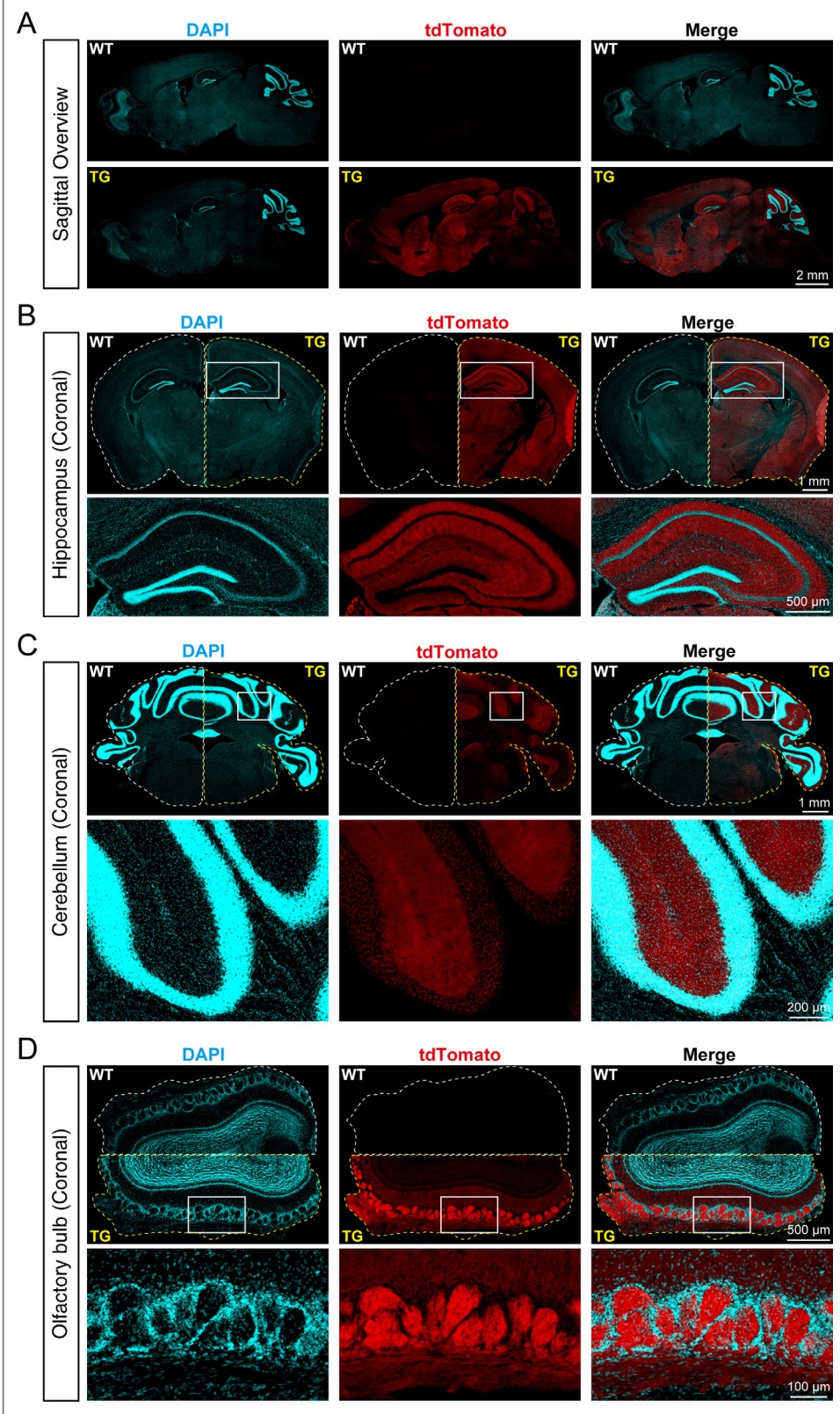

**Figure 2.** tdTomato-fused synaptotagmin-1 (SYT1-tdTomato) is expressed in different brain regions of transgenic mice. (**A**) Representative fluorescence images of the sagittal plane of the brains in *Syt1*-tdTomato transgenic mice and wild-type (WT) mice (8 weeks). The SYT1-tdTomato signals were shown in channel tdTomato (red). All nuclei were stained with DAPI (cyan). (**B–D**) Representative fluorescent images in different brain areas were obtained

*Figure 2 continued on next page*

*Figure 2 continued*

from the *Syt1*-tdTomato transgenic mice and WT mice (8 weeks). (**B**) Hippocampus and cortex; (**C**) cerebellum; (**D**) olfactory bulb.

on the cultured hippocampal neurons from *Syt1*-tdTomato transgenic mice and WT mice. We recorded the action potentials in these neurons and found that the amplitude, frequency, and half-width of spontaneous action potentials in the neurons of *Syt1*-tdTomato transgenic mice are not significantly different with the WT mice (Amplitude: WT: 84.32 ± 2.930 pA, TG: 78.99 ± 3.325 pA; Frequency: WT: 0.4843 ± 0.1379 Hz, TG: 0.5424 ± 0.1391 Hz; Half width: WT: 2.178 ± 0.2616 ms, TG: 2.301 ± 0.2123 ms) (*Figure 5A–C*). These results demonstrate that insertion *Syt1*-tdTomato does not affect the excitability of neurons in transgenic mice.

We also examined the excitatory and inhibitory synaptic transmission mediated by AMPA receptors (AMPARs) or GABA receptors (GABARs) by recording the miniature excitatory postsynaptic currents (mEPSCs) and miniature inhibitory postsynaptic currents (mIPSCs). Our results show that both the amplitude and the frequency of mEPSCs and mIPSCs of the *Syt1*-tdTomato transgenic mice neurons are similar to the WT mice (mEPSCs: Amplitude: WT: 21.97 ± 1.949 pA, TG: 22.50 ± 1.670 pA; Frequency: WT: 0.2078 ± 0.06898 Hz, TG: 0.2255 ± 0.05640 Hz; mIPSCs: Amplitude: WT: 28.83 ± 7.330 pA, TG: 31.48 ± 9.982 pA; Frequency: WT: 0.2942 ± 0.02385 Hz, TG: 0.2728 ± 0.03020 Hz) (*Figure 5D–F and G–I*). An analysis of the rising slope and decay time course of the mEPSCs and mIPSCs (*Figure 5F and I*) shows no significant differences between *Syt1*-tdTomato transgenic mice and WT mice, indicating that the expression of SYT1-tdTomato did not influence the kinetics of mEPSCs and mIPSCs. Thus, our results demonstrate that the insertion of *Syt1*-tdTomato does not affect basal synaptic transmission.

## Neurons from *Syt1*- tdTomato transgenic mice can be used for screening CAMs in an artificial synapse formation assay

To test whether *Syt1*-tdTomato transgenic mice could be used for screening CAMs in an artificial synapse formation assay, we performed a co-culture assay between neurons from *Syt1*-tdTomato transgenic mice and HEK293T cells transfected with Neuroligin 2 and green fluorescent protein (GFP). NLGN2 promotes synapse formation in ASF assays which can be used as a positive control (*Scheiffele et al., 2000*; *Sando et al., 2019*). After being co-cultured for 36–48 hr, the cells were stained and observed using confocal microscopy. We observed a strong accumulation of tdTomato signals around the transfected HEK293T cells (indicated by GFP). Additionally, the accumulated signals overlapped highly with the staining signals of SYN1 (*Figure 6A*).

To further confirm that neurons from *Syt1*-tdTomato transgenic mice can be used in ASF assays to screen synaptogenic molecules, we also tested other synaptogenic CAMs in ASF assays using neurons of *Syt1*-tdTomato transgenic mice. Our results showed that NLGN2, LRRTM2, and SLITRK2 enabled the formation of synaptic connections between transfected HEK293T cells and neurons from *Syt1*-tdTomato transgenic mice. We compared the percentage of synapse-positive HEK293T cells calculated by either tdTomato's signals or the stained synapsin signals as previously reported (*Jiang et al., 2017b*), and found that both signals had the ability to label the synapse accumulated around the HEK293T cells (SYN1: Control: 6.802 ± 1.329%, NLGN2: 88.65 ± 1.077%, LRRTM2: 56.69 ± 2.234%, SLITRK2: 37.82 ± 2.216%; tdTomato: Control: 15.70 ± 1.949%, NLGN2: 88.94 ± 0.9419%, LRRTM2: 60.48 ± 2.186%, SLITRK2: 48.47 ± 2.469%) (*Figure 6B and C*). According to the above results, we thought that neurons from *Syt1*-tdTomato transgenic mice are suitable for screening CAMs in an ASF assay.

## Discussion

In our study, we generated transgenic mice by expressing fluorescent proteins to label synapses and skip the immunofluorescence staining step during synaptogenic adhesion molecules screening. We chose SYT1 as a media to transport tdTomato to the synapse by fusing tdTomato to the C-tail of SYT1. We identified the synaptic localization of SYT1-tdTomato and found that overexpression of SYT1-tdTomato has no effect on synaptic transmission in the transgenic mice and neurons. We demonstrated the availability of neurons from *Syt1*-tdTomato transgenic mice for ASF assay, which is useful for synaptogenic adhesion molecule screening.

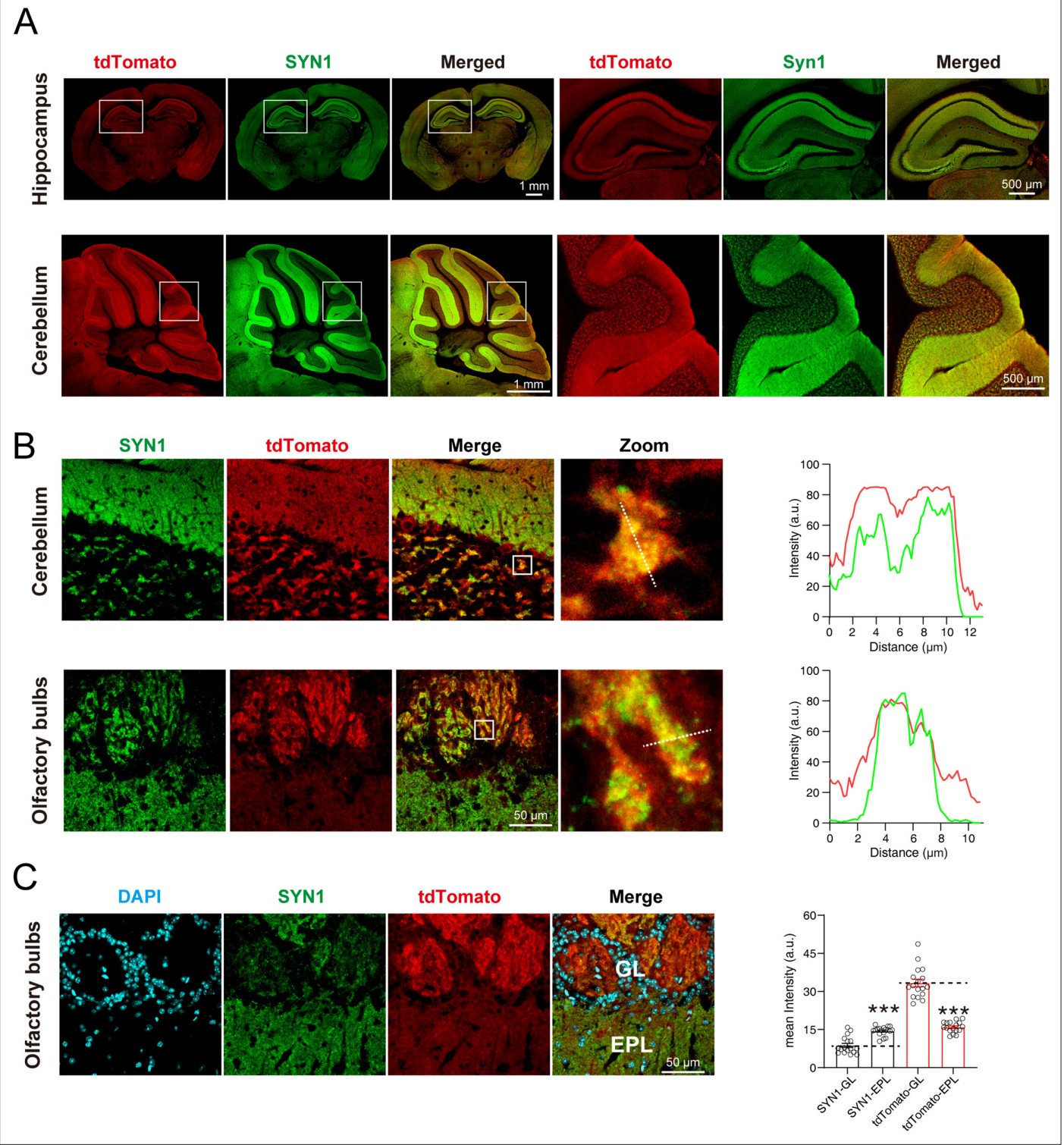

**Figure 3.** tdTomato-fused synaptotagmin-1 (SYT1-tdTomato) is colocalized with synaptic markers in situ. (**A**) The images of SYT1-tdTomato (tdTomato) signals and synapsin1 (SYN1) signals in hippocampus and cerebellum. (**B**) Magnified images of SYT1-tdTomato and SYN1 in granular layer of cerebellum and the glomerular layer of olfactory bulb. The fluorescence intensity profiles of dash lines in zoomed images are plotted on the right. (**C**) The represent images (left) and mean intensity (right) of SYT1-tdTomato and SYN1 in glomerular layer (GL) and external plexiform layer (EPL) of olfactory bulb. All summary graphs show the mean ± standard error of the mean (SEM); statistical analysis was made by t-test (n=17 images in each group, ***p<0.001).

The online version of this article includes the following source data for figure 3:

**Source data 1.** Numerical data corresponding to the graph in panels B and C.

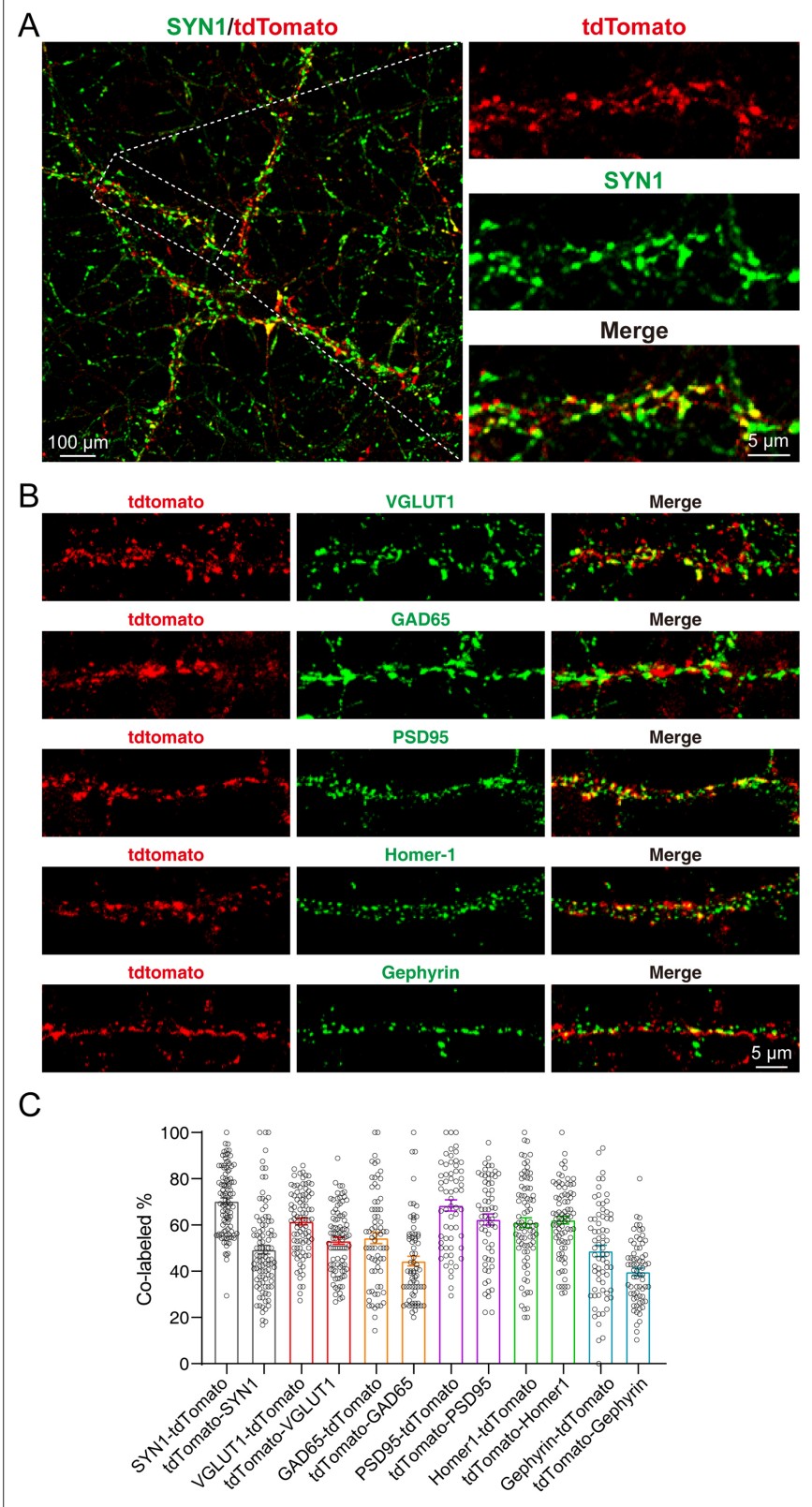

**Figure 4.** tdTomato-fused synaptotagmin-1 (SYT1-tdTomato) signals localize to the synaptic site in cultured neurons. (**A–B**) Representative images of hippocampal neurons from *Syt1*-tdTomato transgenic mice stained with antibodies against SYN1, VGLUT1, GAD65, PSD95, Homer-1, and Gephyrin at DIV 14. (**C**) Summary graphs of co-labeling rate of SYT1-tdTomato with SYN1, VGLUT1, GAD65, PSD95, Homer-1, and Gephyrin (SYN1: n=98/3,

*Figure 4 continued on next page*

*Figure 4 continued*

VGLUT1: n=90/3, GAD65: n=71/3, PSD95: n=61/3, Homer-1: n=81/3, Gephyrin: n=69/3). All summary graphs show the mean ± standard error of the mean (SEM).

The online version of this article includes the following source data for figure 4:

**Source data 1.** Numerical data corresponding to the graph in panel C.

We designed the SYT1-tdTomato expression vector and injected it into the fertilized eggs of C57BL/6 J mice in order to insert the *Syt1*-tdTomato gene into the genome of C57BL/6 J mice to create transgenic fluorescent mice. Due to random insertion, we do not know the actual insertion site and its number. However, according to the immunoblotting of tissue homogenizations, SYT1-tdTomato was detected in the hippocampus, cortex, cerebellum, and olfactory bulb of *Syt1*-tdTomato transgenic mice (*Figure 1D*). There was little SYT1-tdTomato expressed in other tissues although we use *UBC* promoter to drive SYT1-tdTomato expression. There might be a possible explanation: The gene of *Syt1*-tdTomato is inserted into the genome position where the brain-specific enhancer is located or neuron-specific topological chromatin domain (*Jiang et al., 2017a*; *Court et al., 2014*). The microscopic imaging of frozen sections of *Syt1*-tdTomato transgenic mice brain also show that SYT1-tdTomato signals were observed in the hippocampus, cortex, cerebellum, and olfactory bulb (*Figure 2B, C and D*), and the areas of strong signals were in keeping with previous studies on the distribution of SYT1 in the brain (*Fox and Sanes, 2007*; *Berton et al., 1997*). The co-labeling of SYN1 with SYT1-tdTomato signals in the brain slices (*Figure 3A and B*) further confirmed that the fluorescence of SYT1-tdTomato is well localized to synapses in situ. Although SYT1 is thought to be located at the presynaptic site as a fast calcium sensor on synaptic vesicles, it was also reported to show post-synaptic dendritic distribution (*Hussain et al., 2017*), which can explain that SYT1-tdTomato signals do not completely label synaptic sites in cultured neurons (*Figure 4C*).

Overexpression of synaptic proteins were reported to affect the synaptic function. For instance, overexpression of synaptic vesicle protein synapsin caused the decrease in basal transmission and an increase in homosynaptic depression (*Fioravante et al., 2007*). Overexpression of post-synaptic scaffolding protein SAP97 caused the modifications of spine and synapse morphology (*Poglia et al., 2011*). Overexpression of synaptic CAM neuroligins caused the change in excitatory and inhibitory synapse properties (*Chanda et al., 2017*). The C-terminal ECFP Fusion was reported to impair SYT1 function (*Han et al., 2005*). The overexpression of C-terminal fused SYT1 might affect the presynaptic neurotransmitter release. It might further affect the synapse organization as the presynaptic neurotransmitter release can signal the organization of the synapse (*Burlingham et al., 2022*). We found that the electrophysiological properties of *Syt1*-tdTomato transgenic mice neurons were not significantly different from WT mice neurons (*Figure 5*), which might be due to that the expression levels of SYT1-tdTomato in the brain of transgenic mice were far less than the native SYT1 (*Figure 1D*). This provides us with better conditions for synaptogenic adhesion molecules screening with fewer effects on synaptic transmission. This also offers the possibility for *Syt1*-tdTomato transgenic mice to be used for labeling synapses to facilitate the study of electrophysiological recording and imaging in vivo.

Several fluorescently tagged synaptic protein transgenic mice model, such as YFP tagged synaptophysin and pHluorin tagged synaptobrevin have been developed to label synapses (*Umemori et al., 2004*; *Li et al., 2005*). While these models can label synapse well, it lacks the functional analysis of neurotransmitter release in the overexpressed neurons as synaptophysin and synaptobrevin were reported to play a role in regulating neurotransmitter release. Considering the overexpression of synaptobrevin or synaptophysin were reported to promote neurite elongation or enhance neurotransmitter secretion, the synaptic organization and synaptic transmission might be changed in these models. Weiping Han et al. in their previous work (*Han et al., 2005*) have generated transgenic mice expressing a SYT1-ECFP fusion protein. The *Syt1*-ECFP mice expressed the fluorescent protein ECFP in the cortex, midbrain, and cerebellum. However, the expression pattern in their model showed some differences with ours: In the olfactory bulb, the SYT1-tdTomato signals were highly enriched in glomerular layer in our model, which was not observed in the previously reported *Syt1*-ECFP transgenic mice (*Han et al., 2005*). It suggested a potential application of our model in labeling synapses in the glomerular layer of olfactory bulb compared with *Syt1*- ECFP transgenic mice.

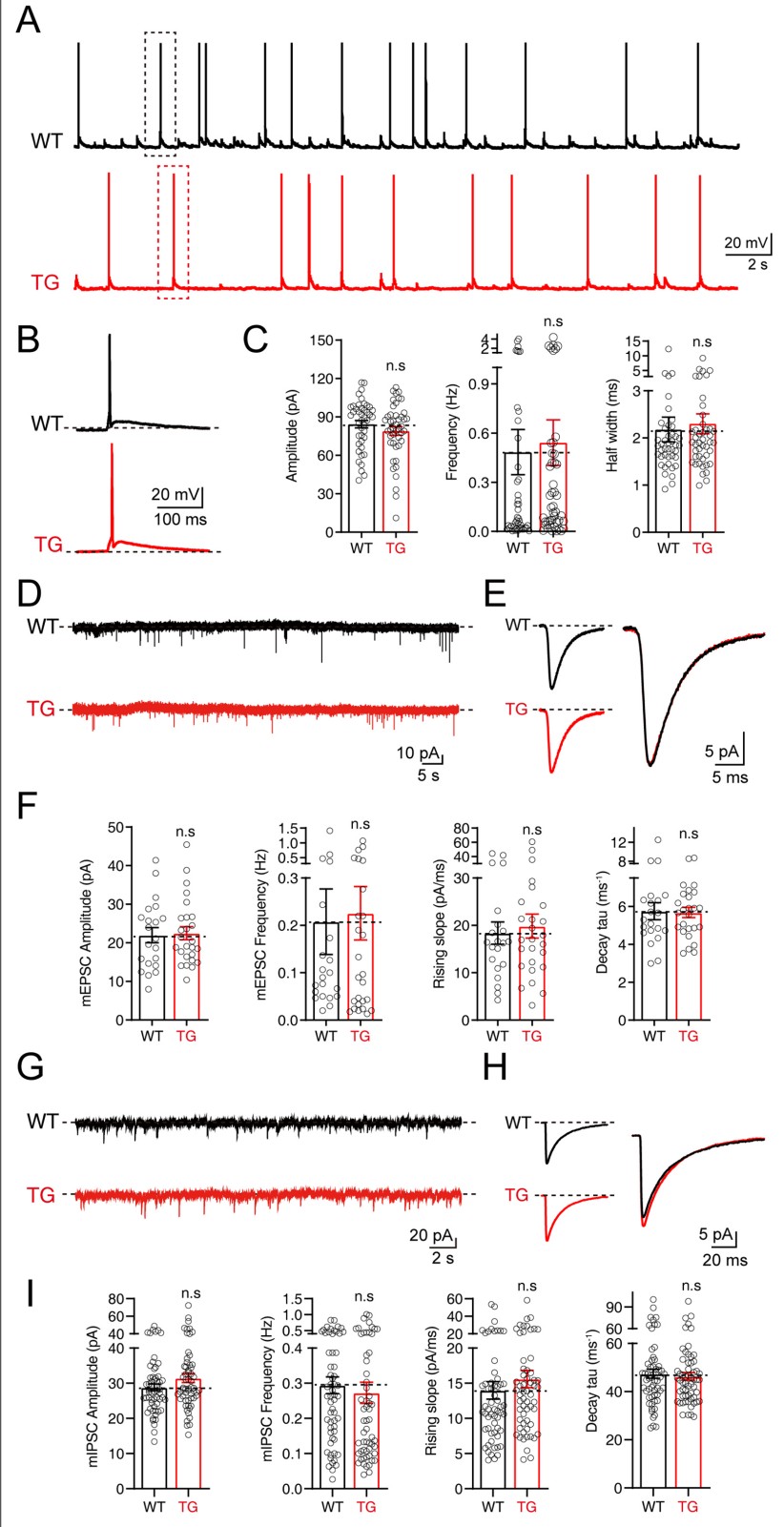

**Figure 5.** No significant changes in electrophysiological properties were observed in neurons of tdTomato-fused synaptotagmin-1 (*Syt1*-tdTomato) transgenic mice. (**A–C**) The spontaneous action potentials were recorded in hippocampal neurons of wild-type (WT) mice and *Syt1*-tdTomato transgenic mice (TG) at DIV 13–15. (**A**) Representative traces of action potential; (**B**) representative traces of single action potential; (**C**) summary

*Figure 5 continued on next page*

*Figure 5 continued*

graphs of the amplitude (left), frequency (center), and half widths (right). (WT: n=43/3; TG: n=45/3, amplitude: p=0.235, frequency: p=0.768, half widths: p=0.716). (**D–F**) The mEPSCs were recorded in hippocampal neurons of WT mice and *Syt1*-tdTomato transgenic mice in 1 µM tetrodotoxin (TTX) and 0.1 mM picrotoxin (PTX) at DIV 13–15. (**D**) Representative traces of mEPSCs; (**E**) representative traces of normalized traces of mEPSCs; (**F**) summary graphs of the frequency (left), amplitude (center left) of mEPSCs and the rising slope (center right), decay $\tau$ (right) of normalized traces of mEPSCs. (WT: n=21/3; TG: n=26/3, amplitude: p=0.837, frequency: p=0.842, rising slope: p=0.681, decay $\tau$: p=0.894). (**G–I**) The mIPSCs recordings in hippocampal neurons of WT mice and *Syt1*-tdTomato transgenic mice in 1 µM TTX and 10 µM CNQX at DIV 13–15. (**G**) Representative traces of mIPSCs; (**H**) representative traces of normalized traces of mIPSCs; (**I**) summary graphs of the frequency (left), amplitude (center left) of mIPSCs and the rising slope (center right), decay $\tau$ (right) of normalized traces of mIPSCs. (WT: n=59/4; TG: n=59/4, amplitude: p=0.102, frequency: p=0.578, rising slope: p=0.355, decay $\tau$: p=0.654). All summary graphs show the mean ± standard error of the mean (SEM); statistical comparisons were made by a two-tailed unpaired t-test (n.s, not significant).

The online version of this article includes the following source data for figure 5:

**Source data 1.** Numerical data corresponding to the graph in panel C, F, I.

In the previous work synaptophysin-YFP was used to label neurites in pontine or vestibular explants-neuronal granule cells co-culture assay (*Umemori et al., 2004*), it provided us with an idea that the synaptic labeling neurons could be used for visualizing artificial synapse formation between neurons and non-neuronal cells. *Syt1*-tdTomato transgenic mice neurons were able to label positive HEK293T cells expressing synaptogenic molecules (NLGN2, LRRTM2, and SLITRK2) without immunostaining (*Figure 6A, B and C*). Because NLGN2, LRRTM2, and SLITRK2 are mainly expressed at the postsynaptic site, they can be recognized by presynaptic terminals marked by SYT1-tdTomato to form intact synapses. Even though it has been suggested that SYT1 is also expressed postsynaptic site, it remains unclear to us whether *Syt1*-tdTomato transgenic mice neurons can be used for screening synaptogenic molecules located at presynaptic site. In order to establish a complete screening system, we have also started to construct the transgenic fluorescent mice model that expresses fluorescent proteins and specifically transports them to postsynaptic spines.

The ASF assays using *Syt1*-tdTomato transgenic mice to screen for synaptogenic molecules have skipped the necessity of immunostaining. However, we still can only test the synapse accumulation around HEK293T cells which are transfected with the same plasmid. It requires large numbers of neurons and HEK293T cells for large-scale screening. To optimize the screening process for large-scale and high-throughput screening, we are examining the co-culture of neurons with multiple HEK293T cells transfected with different plasmids in the same well. The synapse-accumulated HEK293T cells in the co-culture system can be separated by cell digestion, single-cell picking, and single-cell sequencing can be used to identify the genotype of the picked cells. If it works, it will largely improve the throughput of screening in ASF assay.

## Materials and methods
### Construction of *Syt1*-tdTomato expression vector and generation of Syt1-tdTomato transgenic mice

The PiggyBac vector and *UBC* promoter (human ubiquitin C promoter) were chosen to construct the *Syt1*-tdTomato expression vector. In the PiggyBac vector, there are two PiggyBac ITRs located at the sides of the '*UBC* promoter-Kozak-*Syt1*-tdTomato-polyA' cassette in order to promote transposes mediated transgene integration. The constructed vectors were co-injected with transposes into fertilized eggs from C57BL/6 J mice. The offspring were identified by PCR to select those carrying the required PiggyBac transgene. Counter-screening was performed on positive founder mice for transposes. The Cyagen Biosciences (Guangzhou) Inc conducted all of the processes.

### Genotyping

Genomic DNA was isolated from the tailpiece. This was done by taking an appropriate amount of rat tail (about 0.5 cm long) and placing it in a 1.5 mL EP tube. We then added 200 µL 50 mM NaOH and heated in a 100 °C water bath for 30 min for lysis. After vortex mixing, 16 µL 1 M tris-HCL was added

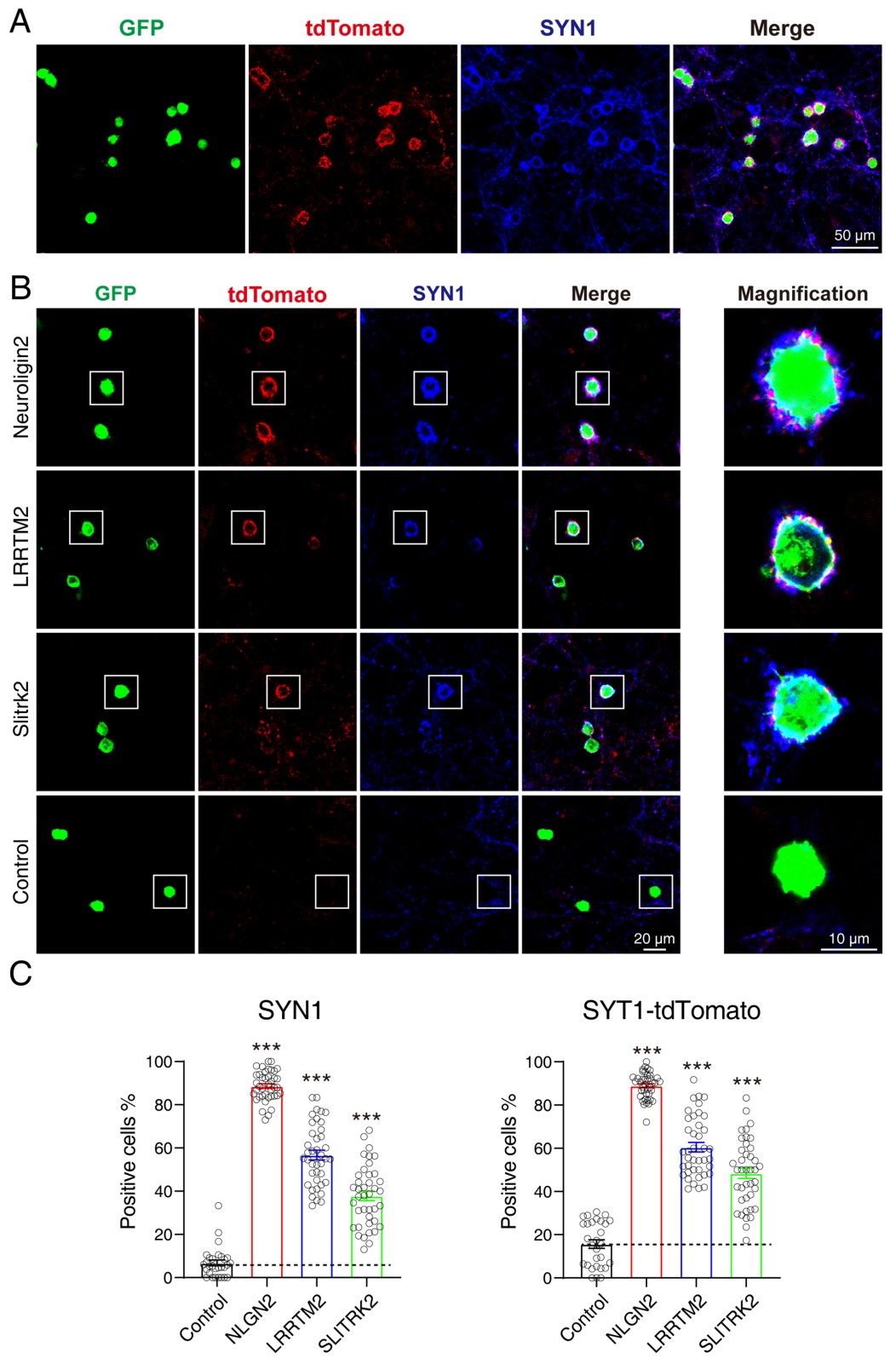

**Figure 6.** tdTomato-fused synaptotagmin-1 *(Syt1*-tdTomato) transgenic mice hippocampal neurons can be used in artificial synapse formation experiments to screen for synaptogenic molecules. (**A**) Representative fluorescent images of co-cultured neurons from *Syt1*-tdTomato transgenic mice with HEK293T cells transfected with plasmid encoding NLGN2 and GFP, which are also stained with synapsin antibody. (**B**) Representative fluorescent images

*Figure 6 continued on next page*

*Figure 6 continued*

of co-cultured neurons from *Syt1*-tdTomato transgenic mice with HEK293T cells transfected with plasmid encoding NLGN2, LRRTM2, SLITRK2, and GFP. (**C**) Summary graphs of the percentage of synapse-positive HEK293T cells from the panel B. (SYN1: Control: n=54 cells out of 854 cells/3 cultures; NLGN2: n=1417 cells out of 1599 cells/3 cultures; LRRTM2: n=692 cells out of 1259 cells/3 cultures; SLITRK2: n=374 cells out of 1032 cells/3 cultures. tdTomato: Control: n=148 cells out of 854 cells/3 cultures; NLGN2: n=1411 cells out of 1599 cells/3 cultures; LRRTM2: n=740 cells out of 1259 cells/3 cultures; SLITRK2: n=486 cells out of 1032 cells/3 cultures). All summary graphs show the mean ± standard error of the mean (SEM); statistical analysis was made by one-way ANOVA (***p<0.001).

The online version of this article includes the following source data for figure 6:

**Source data 1.** Numerical data corresponding to the graph in panel C.

to the end lysis. For PCR analysis, primers were designed according to the *Syt1*-tdTomato expression vector (primer F: TAGGGTAGGCTCTCCTGAATCGAC; primer R: ATTCATAAACTTCTGCTTCAGCTT G). The PCR was performed with the 2 × Taq Plus Master Mix II (Vazyme, P213-01) and according to the manufacturer's instructions. The PCR products carried out electrophoresis on 1.5% agar gel. The amplification product was 396 bp.

## Primary hippocampal neuronal culture

Neurons for experiments were obtained from postnatal day 0 C57BL/6 J mice's hippocampus. The mice's brains were isolated, and the hippocampus were dissected on ice. After washing with DMEM (Gibco, 11995–065), the hippocampus was placed in 0.25% trypsin (Sigma, T4049) and digested at 37 °C for 12 min. We added DMEM to end the digestion, and to count after blowing and mixing. Neurons were plated on glass coverslips (diameter 12 mm) that were coated with poly-D-lysine (Gibco, A38990401) at a density of 80,000 neurons per coverslip and cultured in Neurobasal Plus Medium (Gibco, A35829-01) containing 2% B-27 Plus Supplement (Gibco, A3592801) at 37 °C in 5% $CO_2$ in a thermostatic cell incubator.

## Tissues protein extraction and western blotting

Protein was extracted from tissues of transgenic and wildtype mice using RIPA lysis buffer (Thermo Scientific, 89901) containing protease inhibitors (Roche, 4693132001). The concentration of extracted proteins were measured with BCA protein assay kit (Thermo Scientific, 23235) according to the manufacturer's instructions. SDS-PAGE was performed by using NuPAGE precast gels (Invitrogen, NP0303BOX). The membrane was scanned with an infrared imaging system (Odyssey). Antibodies against tdTomato (Novus, NBP2-78136, 1:1000 dilution) and vasolin-containing protein (ABclonal, A13368, 1:1000 dilution) were used.

## Immunofluorescence staining of neurons

The cultures were fixed with 4% paraformaldehyde in phosphate-buffered saline (PBS, pH 7.4) for 10 min. After washing with PBS, the cultures were blocked and permeabilized with Blocking Buffer (0.3% Tween 20, 5% skimmed milk, and 2% goat serum in PBS). Cultures were incubated overnight with primary antibodies [anti-SYN1 (1:20,000; homemade), anti-VGLUT1 (1:2000; Synaptic Systems, 135 302), anti-GAD65 (1:5000; Sigma, G5638), anti-PSD95 (1:2000; NeuroMab, 75–028), anti-Homer-1 (1:1000; Synaptic Systems, 160,011), or anti-Gephyrin (1:1000; Synaptic Systems, 147 011)] diluted in Blocking Buffer. After washing three times for 5 min in PBS, cultures were incubated for 30 min at room temperature with secondary antibodies [Goat anti-Rabbit IgG (H+L) Highly Cross-Adsorbed Secondary Antibody, Alexa Fluor Plus 647 (1:200; Invitrogen, A32733) or Dylight 649, Goat Anti-Mouse IgG (1:200; Abbkine, A23610)]. Samples were placed in mounting medium (Southernbiotech, 0100–01) after washing them three times for 5 min in PBS and once in double distilled $H_2O$ (dd$H_2O$).

## Immunofluorescence staining of mouse brain slices

Adult mice (transgenic and WT) were deeply anesthetized with tribromoethanol and then transcardially perfused with 4% paraformaldehyde in phosphate-buffered saline (PBS, pH 7.4). The brains were removed and fixed in the 4% paraformaldehyde in PBS for 24 hr. After tissue dehydration with 15% and 30% sucrose solution in PBS, brains were embedded with a freezing embedding medium

(optimum cutting temperature compound). The 40 µm thick sections were cut on a cryo-cut cryostat microtome. The sections were permeabilized with 0.3% Triton X-100 in PBS. After washing for 5 min in PBS, sections were incubated in PBS containing 5 µg/mL DAPI (Beyotime, C1002) for 20 min in order to stain cell nuclei, washed with PBS, and placed in the mounting medium.

### Electrophysiological recordings

Neurons at DIV 13–15 were taken for the experiment. Current clamp recordings of primary hippocampal neurons were carried out with a MultiClamp 700 A amplifier (Molecular Devices). Series resistance was compensated to 60–70%, and recordings with series resistances of >20 MΩ were rejected. For AP recordings, neurons were patched and held in the current-clamp whole-cell configuration, and maintained in an external solution of 150 mM NaCl, 4 mM KCl, 10 mM HEPES, 2 mM $CaCl_2$, 1 mM $MgCl_2$, 10 mM glucose (pH 7.40, Osm 315 mOsm/kg). Microelectrodes (World Precision Instruments) were filled with internal solution, which contains 145 mM KCl, 5 mM NaCl, 10 mM HEPES, 5 mM EGTA, 4 mM MgATP, 0.3 mM $Na_2GTP$ (pH 7.25, Osm 305 mOsm/kg). Whole-cell voltage-clamp recordings of primary hippocampal neurons were carried out using a MultiClamp 700 A amplifier (Molecular Devices). Coverslips, seeded with primary hippocampal neurons, were kept in an external solution of 150 mM NaCl, 4 mM KCl, 2 mM $CaCl_2$, 1 mM $MgCl_2$, 10 mM HEPES, and 10 mM Glucose (pH 7.40, Osm 315 mOsm/kg). For mEPSC recordings, Tetrodotoxin (1 µM) was added to the external solution to block evoked synaptic responses. Patch recording pipettes (3–6 MΩ, World Precision Instruments) were filled with an internal solution of 110 mM Cs- Methanesulfonate, 20 mM TEA-Cl, 8 mM KCl, 10 mM EGTA, 10 mM HEPES, 3 mM MgATP, and 0.3 mM $Na_2GTP$ (pH 7.3, Osm 275–285 mOsm/kg). For mIPSC recordings, patch recording pipettes (3–6 MΩ) are filled with an internal solution containing the following (in mM): 140 mM CsCl, 10 mM HEPES, 2 mM $MgCl_2$, 4 mM $Na_2ATP$, 0.4 mM $Na_3GTP$, and 5 mM EGTA (pH 7.3, Osm 275–285 mOsm/kg). Tetrodotoxin (1 µM), NBQX (10 µM), and D-APV (50 µM) was added to the external solution. The data were analyzed with Clampfit 10.2 (pClamp) and Igor 4.0 (WaveMetrics).

### HEK293T cell culture and transfection

HEK293T cells (KCB 200971YJ) were obtained from the Kunming Cell Bank, Chinese Academy of Sciences. The cell line was authenticated by STR profiling by the supplier and was checked for mycoplasma contamination regularly. They were cultured in COS medium (DMEM containing 10% FBS, 50 U/mL penicillin, 50 µg/mL streptomycin, 44.52 mM $NaHCO_3$, pH 7.2–7.4) at 37 °C in 5% $CO_2$ in a thermostatic cell incubator. For passaging, when cells grow to 80% cell density, discard the old medium, and add PBS to wash away metabolic waste and floating dead cells. Add 0.05% of Trypsin solution, digest in the incubator for 2 min, and add 2 times the volume of COS medium to terminate the digestion. Centrifuge the cell suspension and add a COS medium to resuspend the cells to culture. The HEK293T cells to be transfected were cultured in 24-well plates. After the cell density grew to about 80%, transfection was performed. Add plasmid and polyethyleneimine (PEI; 1 mg/mL in $ddH_2O$) to 35 µl OPTI. The PEI plasmid ratio was 3:1. The mixture was incubated at room temperature for 30 min and then added dropwise to HEK293T cell cultures. For one well of a 24-well plate, 1.2 µg of each plasmid and 0.3 µg of GFP was transfected.

### Artificial synapse formation

After 24 hr of transfection, the HEK293T cells were digested by adding 0.05% trypsin, placed at 37°C for 1 min, and the digestion was terminated by adding a COS medium. The cell suspension was centrifuged, resuspended in Neurobasal Plus Medium (containing B-27), counted, and then added dropwise into hippocampal neuronal culture at DIV 9 in vitro (40,000/coverslip), gently shaken, and placed at 37 °C in 5% $CO_2$ in a thermostatic cell incubator. After 36–48 hr of incubation, immunocytochemical assays were performed.

### Confocal imaging and image analysis

For immunofluorescence staining of neurons and mouse brain slices, images were captured using a confocal microscope (Olympus FV3000). Image J was used to analyze the co-labeling rate, and Matlab was used to analyze positive cells in the artificial synapse formation assay.

## Statistical analysis

Statistical analyses were conducted using GraphPad Prism 8.0.1 software for t-test. Results were displayed as mean ± standard error (SEM). Differences in means were accepted as significant if the results of the two-tailed unpaired t-test and one-way ANOVA were $p < 0.05$.

## Acknowledgements

This work was supported by grants from the National Natural Science Foundation of China (81925011, 92149304, 32170954, 32100763); Key-Area Research and Development Program of Guangdong Province (2019B030335001); Beijing Advanced Innovation Center for Big Data-based Precision Medicine, Capital Medical University, Beijing, China (PXM2021_014226_000026).

## Additional information

### Funding

| Funder | Grant reference number | Author |
| --- | --- | --- |
| National Natural Science Foundation of China | 81925011 | Chen Zhang |
| National Natural Science Foundation of China | 92149304 | Chen Zhang |
| National Natural Science Foundation of China | 32170954 | Mengping Wei |
| National Natural Science Foundation of China | 32100763 | Lei Yang |
| Key Area Research and Development Program of Guangdong Province | 2019B030335001 | Chen Zhang |
| Capital Medical University | PXM2021_014226_000026 | Chen Zhang |

The funders had no role in study design, data collection and interpretation, or the decision to submit the work for publication.

### Author contributions

Lei Yang, Data curation, Formal analysis, Validation, Investigation, Visualization, Methodology, Writing - original draft; Jingtao Zhang, Data curation, Investigation, Visualization, Methodology, Writing - original draft; Sen Liu, Data curation, Formal analysis, Methodology; Yanning Zhang, Data curation, Investigation, Methodology; Li Wang, Investigation, Methodology; Xiaotong Wang, Shanshan Wang, Ke Li, Methodology; Mengping Wei, Supervision, Funding acquisition, Validation, Writing - original draft, Project administration, Writing - review and editing; Chen Zhang, Conceptualization, Resources, Supervision, Funding acquisition, Validation, Writing - original draft, Project administration, Writing - review and editing

### Author ORCIDs

Lei Yang ⓘ http://orcid.org/0000-0002-2882-5560
Chen Zhang ⓘ http://orcid.org/0000-0002-7940-8054

### Ethics

This study used C57BL/6J WT mice and Syt1-tdTomato transgenic mice (P0-P56). Animals were housed at room temperature (RT) 20 ± 2°C, with a 12-hour light-dark cycle, air circulating, and unrestricted access to food and water. All animal studies were conducted according to the Guide for the Care and Use of Laboratory Animals (8th edition) and approved by the Animal Experiments and Experimental Animal Welfare Committee of Capital Medical University (Approval ID: AEEI-2019-013).

### Decision letter and Author response

Decision letter https://doi.org/10.7554/eLife.81884.sa1

Author response https://doi.org/10.7554/eLife.81884.sa2

## Additional files

### Supplementary files
• MDAR checklist

### Data availability
All data generated or analysed during this study are included in the manuscript and supporting files. Source data have been provided for Figures 1–6.

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
