## [Editor Report]

In this manuscript, Zhang and colleagues created a transgenic mouse strain that express SYT-1-tdt in all neurons. They showed that the labelled SYT-1 does not represent a strong over expression, colocalizes with multiple synaptic markers and label synapses in different regions. Importantly, they showed that the transgenic expression does not alter synaptic function using electrophysiogical assays. This reagent can be used to visualize synapses in vivo and in cultures.

---

## [Decision Letter]

**Decision letter after peer review:**

Thank you for submitting your article "Establishment of Transgenic Fluorescent Mice for Screening Synaptogenic Adhesion Molecules" for consideration by *eLife*. Your article has been reviewed by 3 peer reviewers, one of whom is a member of our Board of Reviewing Editors, and the evaluation has been overseen by John Huguenard as the Senior Editor. The reviewers have opted to remain anonymous.

Essential revisions:

1) To provide a further comparison between this transgenic strain with other existing strains in the literature.

2) To improve the impact of this manuscript by demonstrating more utility of the transgenic strain.

*Reviewer #1 (Recommendations for the authors):*

The authors should test to what extent syt-2-tdt is expressed compared to the endogenous syt-1 level. This can be achieved with a western blot.

It seems that this mouse strain will be more useful than simply being used for synaptogenic screens in cultures. If that is the only use, I would think that the impact is not very high. I think it can also be used to label synapses in situ without immunostaining.

*Reviewer #2 (Recommendations for the authors):*

Yang et al. designed and generated Syt1-TDT transgenic mice, and analyzed them in cultured neurons and brain sections, showing that Syt1-TDT immunofluorescent signals colocalize with known synaptic marker proteins. The authors then showed extensive electrophysiological recordings showing that neurons from Syt1-TDT transgenic mice were electrophysiologically comparable to control neurons. Lastly, they performed artificial synapse formation assays showing that Syt1-TDT signals are recruited to HEK293T cells expressing known synaptogenic adhesion proteins.

I agree with the authors that this new mouse line could be potentially useful for various cell biological analyses in neurons. However, to me, the advantage of this mouse line lies only in skipping the immunostaining procedures. Moreover, Syt1-TDT signals do not completely label synaptic sites in cultured neurons, based on colocalization % values (Figure 2).

Another concern is that the Introduction part of this manuscript was not written well to cover major progress in the field of mammalian synaptic adhesion proteins. The authors should cite more relevant research articles and a series of key review papers in the field. I recommend the authors that they should look up the following terms and cite a subset of references at appropriate places of text: synapse formation, synaptic adhesion, and synaptogenic proteins.

*Reviewer #3 (Recommendations for the authors):*

While this mouse model will undoubtedly be useful, it is less clear whether this study presents a significant conceptual advance as several other mouse models expressing fluorescently tagged synaptic vesicle proteins have been available from the Jackson Labs. These current models include a similar model expressing synaptotagmin 1-eCFP fusion protein (B6SJL-Tg(Thy1-Syt1/ECFP)1Sud/J) or mice expressing a synaptophysin-YFP transgene (B6.Cg-Tg(Thy1-YFP/Syp)10Jrs/J). Moreover, spH21 transgenic mice (B6;CBA-Tg(Thy1-spH)21Vnmu/J) generated by the Murthy lab expressing fluorescent indicator synaptopHluorin (synaptobrevin and pHluorin chimera) have been characterized and available. Expression of the pHluorin tag provides the added advantage that one can test the presynaptic functionality of boutons formed in co-culture assays.

---

## [Author Response]

Essential revisions:1) To provide a further comparison between this transgenic strain with other existing strains in the literature.2) To improve the impact of this manuscript by demonstrating more utility of the transgenic strain.Reviewer #1 (Recommendations for the authors):The authors should test to what extent syt-2-tdt is expressed compared to the endogenous syt-1 level. This can be achieved with a western blot.

We thank the reviewer for raising this point. To test the extent of expressed Syt1-TDT in the transgenic mice, we performed the western blot using antibody against Syt1 in the sample of WT and transgenic mice in the revised manuscript. We found that immunoblotting using antibody against tdTomato detected a band at near 115 kd, which is the molecular weight of Syt1-TDT, in the transgenic mice but not WT mice (Figure 1C). And mmunoblotting using antibody against Syt1 detected the bands at 115 kd and below 65 kd (the molecular weight of native Syt1) in hippocampus, cortex, cerebellum, and olfactory bulb (Figure 1D). We found that the intensity of the band of Syt1 is far more than Syt1-TDT, which suggested that the expresion level of Syt1-TDT is much lower than the native Syt1.

It seems that this mouse strain will be more useful than simply being used for synaptogenic screens in cultures. If that is the only use, I would think that the impact is not very high. I think it can also be used to label synapses in situ without immunostaining.

We thank the reviewer for the suggestion. To verify the synaptic labeling of Syt1-TDT in situ, we performed additional experiments to label the synapse by antibody against synapsin in brain slice of Syt1-TDT transgenic mice. We found that Syt1-TD showed same distribution with synapsin in hippocampus and cerebellum (Figure 3A). In the image of higher magnification, the Syt1-TDT were highly overlapped with synapsin in the granular layer of cerebellum and glomerular layer of olfactory bulb (Figure 3B). These result suggested that Syt1-TDT can be used to label synapses in situ without immunostaining in the transgenic mice.

Reviewer #2 (Recommendations for the authors):Yang et al. designed and generated Syt1-TDT transgenic mice, and analyzed them in cultured neurons and brain sections, showing that Syt1-TDT immunofluorescent signals colocalize with known synaptic marker proteins. The authors then showed extensive electrophysiological recordings showing that neurons from Syt1-TDT transgenic mice were electrophysiologically comparable to control neurons. Lastly, they performed artificial synapse formation assays showing that Syt1-TDT signals are recruited to HEK293T cells expressing known synaptogenic adhesion proteins.I agree with the authors that this new mouse line could be potentially useful for various cell biological analyses in neurons. However, to me, the advantage of this mouse line lies only in skipping the immunostaining procedures. Moreover, Syt1-TDT signals do not completely label synaptic sites in cultured neurons, based on colocalization % values (Figure 2).

We thank the reviewer for raising these points. As mentioned above, we performed additonal experiments to verify the usage of our model in labeling synapses in situ in the revised manuscript (Figure 3). In the ASF assay, the advantage of our model indeed lies in skipping the immunostaining procedures, however, we thought it is critical for the large-scaled CAMs screening by ASF assay. First, the large-scaled screening will cost a lot of time and work in immnuostaining procedure. The cost of antibodies is also a huge burden. Second, we discussed the possibility of optimizing the screening process in the section of discussion: We are examining the co-culture of neurons with multiple HEK293T cells with different plasmids overexpressed in the same well. The synapse-accumulated HEK293T cells in the co-culture system can be separated by cell digestion, single-cell picking, and single cell sequencing can be used to identify the genotype the picked cells. If it works, it will largely improve the throughput of screening in ASF assay.

As the reviewer mentioned, the Syt1-TDT signals do not completely overlapped with synaptic markers in cultured neurons in the Figure 4C of revised manuscript. We discussed this result in line 264 to 267 in the section of discussion in the revised manuscript:

“Although Syt1 is thought to be located at presynaptic site as a fast calcium sensor on synaptic vesicles, it was also reported to show postsynaptic dendritic distribution [43] , which can explain that Syt1-TDT signals do not completely label synaptic sites in cultured neurons (Figure 4C).”

Another concern is that the Introduction part of this manuscript was not written well to cover major progress in the field of mammalian synaptic adhesion proteins. The authors should cite more relevant research articles and a series of key review papers in the field. I recommend the authors that they should look up the following terms and cite a subset of references at appropriate places of text: synapse formation, synaptic adhesion, and synaptogenic proteins.

We thank the reviewer for pointing out this weakness. We reorganized the part of introduction of CAMs in synapse formation and cite more relevant research and review articles in the line 67 to 80 in the revised manuscript:

“Synaptic cell adhesion molecules (CAMs) are molecules that act as a key role in initiating synapse formation through trans-synaptic interactions, and were originally proposed to enable mechanistic cell-cell recognition [7-9]. A variety of CAMs have been shown to initiate synapse formation, anchor and organize the precise alignment of the pre- and postsynaptic sides of a synapse, and enable enhanced short- and long-term synaptic plasticity of synaptic transmission [1, 10-22]. However, the synapses show diversities in types of neurotransmitter types, release probability, composition of postsynaptic receptor in different brain area, and are organized by specific CAMs [18, 23-25]. Deletion of some CAMs only affect the pre- or post-synaptic organization rather than affect synapse numbers [26, 27]. In other cases, deletion of specific CAMs only induces the loss of synapse number in specific type of neurons or brain area [28, 29]. Therefore, how the CAMs determine specific synapse formation still need to be elucidated and it cannot be excluded that some other CAMs driving synapse formation remain undiscovered.”

To make this part more concise, we deleted the description of specific CAMs’ function in synapse formation but added more description about the limitations of previous findings on CAMs.

Reviewer #3 (Recommendations for the authors):While this mouse model will undoubtedly be useful, it is less clear whether this study presents a significant conceptual advance as several other mouse models expressing fluorescently tagged synaptic vesicle proteins have been available from the Jackson Labs. These current models include a similar model expressing synaptotagmin 1-eCFP fusion protein (B6SJL-Tg(Thy1-Syt1/ECFP)1Sud/J) or mice expressing a synaptophysin-YFP transgene (B6.Cg-Tg(Thy1-YFP/Syp)10Jrs/J). Moreover, spH21 transgenic mice (B6;CBA-Tg(Thy1-spH)21Vnmu/J) generated by the Murthy lab expressing fluorescent indicator synaptopHluorin (synaptobrevin and pHluorin chimera) have been characterized and available. Expression of the pHluorin tag provides the added advantage that one can test the presynaptic functionality of boutons formed in co-culture assays.

We thank the reviewer for raising this point. We discussed the comparation of our transgenic model with the existed ones in line 285 to 300 in the section of discussion in the revised manuscript. Compared with the synaptophysin-YFP and synaptopHluorin mice (Umemori et al. 2004; Li et al. 2005), we tested the electrophysiological properties and synaptic neurotransmission of neurons in transgenic mice. We excluded the effect of Syt1-TDT overepression on synaptic function, which might lead to changed synaptic organization or synaptic transmission. We also compared our model with synaptotagmin 1-eCFP mice, we found that in our model the expression level of Syt1-TDT was far less than the native Syt1, while the expression level of Syt1-ECFP was comparable to the native Syt1 (Han et al. 2005). We thought it might reduce the side effect of Syt1-TDT overexpression. Besides, we found that Syt1-TDT was enriched in the glomerular layer of olfactory bulb compared with Syt1-ECFP transgenic mice. It suggested a potential application of our model in labeling synapse in glomerular layer of olfactory bulb in vivo.

Reference

Han, W., J. S. Rhee, A. Maximov, W. Lin, R. E. Hammer, C. Rosenmund, and T. C. Sudhof. 2005. 'C-terminal ECFP fusion impairs synaptotagmin 1 function: crowding out synaptotagmin 1', *J Biol Chem*, 280: 5089-100.

Li, Z., J. Burrone, W. J. Tyler, K. N. Hartman, D. F. Albeanu, and V. N. Murthy. 2005. 'Synaptic vesicle recycling studied in transgenic mice expressing synaptopHluorin', *Proc Natl Acad Sci U S A*, 102: 6131-6.

Umemori, H., M. W. Linhoff, D. M. Ornitz, and J. R. Sanes. 2004. 'FGF22 and its close relatives are presynaptic organizing molecules in the mammalian brain', *Cell*, 118: 257-70.